# Trends in COVID-19 Vaccine Development: Vaccine Platform, Developer, and Nationality

**DOI:** 10.3390/vaccines12030259

**Published:** 2024-03-01

**Authors:** Ryo Okuyama

**Affiliations:** College of International Management, Ritsumeikan Asia Pacific University, Beppu 874-8577, Japan; ryooku@apu.ac.jp

**Keywords:** COVID-19, vaccine platform, developer, national innovation system

## Abstract

Various vaccine platforms, including emerging platforms, have been applied in the development of COVID-19 vaccines. Biotechnology startups often lead the development of new medical technologies, whereas major pharmaceutical companies and public institutions have long contributed to vaccine development. In this study, vaccine platforms and developers involved in COVID-19 vaccine development were analyzed, elucidating the trends of vaccine platforms used, the country distribution of the developers, and differences in the profiles of developers by vaccine platform technologies and country. The analysis revealed that conventional, established, and emerging vaccine platforms have been widely used and that older platforms are more advanced in clinical development. It also demonstrated the emergence of China, in addition to the U.S., while many pharmerging countries have been engaged in development. Startups have significantly contributed to the development of viral vector and RNA-based vaccines, suggesting their important role in the application of novel technologies. The major developers differ by country and region. Alliances, including international collaborations, have progressed in late clinical development. Based on these results, future perspectives of pandemic vaccine development and implications for policy and corporate strategies are discussed.

## 1. Introduction

Severe acute respiratory syndrome coronavirus 2 (SARS-CoV-2), which emerged at the end of 2019, prompted rapid vaccine development worldwide. The most rapidly developed and globally used vaccines for the coronavirus disease 2019 (COVID-19) use RNA-based or adenoviral vector vaccine platforms, both of which are emerging technologies that have not been or rarely been applied to approved vaccines [1]. In addition, various types of vaccine platforms, including inactivated/live-attenuated virus, protein subunit, virus-like particle, and DNA-based, have been used in approved and clinically developed COVID-19 vaccines [1]. Inactivated and live-attenuated virus vaccine platforms are the most conventional vaccine technologies and have long been applied to many vaccines [2]. A protein subunit vaccine platform was established, and vaccines using this platform have been approved for various infectious pathogens, such as Hepatitis B virus, papillomavirus, and influenza virus, since 1986 [3]. The virus-like particle vaccine platform was established and applied to approved vaccines against the Hepatitis B virus, human papillomavirus, and Hepatitis E virus in the late 1980s [4,5]. Compared with these vaccine platforms, viral vector, RNA-based, and DNA-based vaccine platforms are novel technologies. The first adenoviral vector vaccine approved was the Ebola virus vaccine in 2019, followed by Vaxzevria, the COVID-19 vaccine [6]. Spikevax and Comirnaty, both of which received emergency use authorization for COVID-19 from the U.S. Food and Drug Administration (FDA) in December 2020, were the first examples of RNA-based vaccines in clinical use [7]. To date, only one DNA vaccine (ZyCoV-D) has been authorized for emergency use against COVID-19 in India [8]. A significant advancement in vaccine development was the availability of these emerging vaccine platforms for clinical use through the development of COVID-19 vaccines. Various vaccine platforms have been widely used for COVID-19 vaccine development, including conventional (inactivated/live-attenuated virus), established (protein subunit and virus-like particle), and emerging (viral vector, RNA-based, and DNA-based) ones. Therefore, a comprehensive analysis of the global development status of COVID-19 vaccines could provide an opportunity to gain an overarching understanding of the utilization of vaccine platform technologies worldwide.

A previous study revealed that startup ecosystems, including startups, universities, public institutions, incumbent companies, and investors, played a key role in the rapid development and commercialization of Spikevax, Comirnaty, and Vaxzevria [1]. Young, small biotechnology companies, many of which are university startups, have driven innovative new drug discovery [9,10] and the development of novel drug modality technologies has often been pioneered by biotechnology startups [11,12]. Recently, small companies have expanded their roles to late-entry drug discovery [13]. However, historically, government and public institutions have greatly contributed to vaccine development [14,15]. Large pharma companies, especially the “big four” vaccine makers, including GlaxoSmithKline (Brentford, UK), Merck & Co. (Rahway, NJ, USA), Pfizer (New York, NY, USA), and Sanofi Pasteur (Lyon, France), have been actively engaged in vaccine development [16]. Considering the above, it is speculated that various players may contribute to vaccine development across different vaccine platforms: conventional (inactivated/live-attenuated virus), established (protein subunit and virus-like particle), and emerging (viral vector, RNA-based, and DNA-based). This study elucidates the current global landscape of vaccine developers in COVID-19 vaccines by vaccine platform. The characteristics of vaccine developers by country (country or territory) are also important, with each country having its own national innovation system and national-level innovation processes involving the government, public institutions, and enterprises [17]. Uncovering the profiles of vaccine developers and their national differences will contribute to discussions on the technology and alliance strategies of vaccine-developing companies and the vaccine development promotion measures of the government.

This study uses the term “pharmerging”. A pharmerging country is one where the pharmaceutical market was small but has grown in recent years [18]. This concept also covers the drug development capabilities of the country [19]. In this study, all countries, except the countries/regions with advanced pharmaceutical industries, such as the U.S., Europe, and Japan, are referred to as pharmerging countries. Countries that develop vaccines have drug development capabilities and pharmaceutical markets to some extent; therefore, they can be categorized as pharmerging. China is considered to be a pharmerging country in general, but is not included in the definition used in this study because the presence of China was significant in COVID-19 vaccine development.

## 2. Materials and Methods

### 2.1. Identification of Approved COVID-19 Vaccines and COVID-19 Vaccine Candidates under Clinical Development: Vaccine Platforms, Development Stages, and Developers

The World Health Organization (WHO) compiled detailed information on approved COVID-19 vaccines and COVID-19 vaccine candidates under development as of 30 March 2023 (https://www.who.int/teams/blueprint/covid-19/covid-19-vaccine-tracker-and-landscape, accessed on 21 January 2024). According to this database, 183 COVID-19 vaccines and vaccine candidates were in the clinical phase. Information on these 183 COVID-19 vaccines and vaccine candidates was analyzed in this study. The database includes the vaccine platform used, developers, and the clinical development stage of each vaccine and vaccine candidate. Because the clinical development stage of one vaccine candidate (COVID-19 Oral Vaccine Consisting of Bacillus Subtilis Spores developed by DreamTec Research Limited) was not shown, this vaccine candidate was excluded from the analysis. The developer of one vaccine candidate (Han Xu, M.D., PhD., FAPCR, Sponsor-Investigator, IRB Chair as a developer of the Ad26.cov2.s+bcg vaccine, AD26-BCG) was not the organization but the individual’s name; therefore, this vaccine candidate was excluded from the analysis. One vaccine (ChAdOx1-S-(AZD1222), Covishield, Vaxzevria) was in phases 4 and 1; therefore, it was double-counted during launch/late development (phases 2/3, 3, and 4) and early development (phases 1, 1/2, and 2).

### 2.2. Identification of Developer Nationalities

The country where the headquarters of the company was located if the developer was a company and the institution location if the developer was a university, public institution, or hospital were identified by searching for the name of the organization on the World Wide Web. The distribution of the countries where each developer was located was analyzed, including the U.S., China, Europe (Belgium, France, Italy, the Netherlands, Norway, Spain, Germany, UK, and Sweden), East Asia (Japan, Korea, and Taiwan), Southeast Asia (Indonesia, Singapore, Thailand, and Vietnam), South/Middle East/West Asia (India, Iran, Turkey, Kazakhstan, Saudi Arabia, and Israel), Australia, Latin America (Argentina, Brazil, Cuba, and Mexico), and others (Canada, Egypt, Russia, and Uganda). Then, the vaccine platform used by each developer was identified from the above database and classified by countries/regions.

### 2.3. Identification of Developer Types

The developers were classified into company and university/public institution/hospital (shown as U/P/H in the graphs and Table 2), and the companies were further classified into small- and medium-sized enterprises (SMEs) founded in or after 2000, SMEs founded before 2000, and large enterprises (LEs). The founding year of the company was identified through a web search (mainly from CB Insights (https://www.cbinsights.com/, accessed on 20 January 2024)). The top 50 pharmaceutical companies that demonstrated the highest revenue in 2019, a year before SARS-CoV-2 became globally distributed, were identified on the Top 50 Global Pharma Companies 2019 website (https://www.rankingthebrands.com/The-Brand-Rankings.aspx?rankingID=370&nav=industry, accessed on 20 January 2024). If a developer was among the top 50 companies, the company was classified as an LE. Otherwise, the company was classified as an SME. A similar classification was applied to previous studies [10,13]. The types of developers were analyzed by vaccine platform (Figure 3) and by country/region (Figure 4).

### 2.4. Analysis of Alliance Situation

If a vaccine or vaccine candidate had more than one developer, the vaccine or vaccine candidate was classified as “Alliance”. If all the developers of the “Alliance” vaccine or vaccine candidate were from the same country, the vaccine or vaccine candidate was classified as “Domestic”. Otherwise, the “Alliance” vaccine or vaccine candidate was classified as “International”. “Alliance” vaccines or vaccine candidates were further classified as follows: If developers were all SMEs, the vaccine or vaccine candidate was classified as “SME & SME”. If developers were all LEs, the vaccine or vaccine candidate was classified as “LE & LE”. If developers were all universities/public institutions/hospitals, the vaccine or vaccine candidate was classified as “U/P/H & U/P/H”. If developers included SME and LE, the vaccine or vaccine candidate was classified as “SME & LE”. If developers included SMEs and universities/public institutions/hospitals, the vaccine or vaccine candidate was classified as “SME & U/P/H”. If developers included LEs and universities/public institutions/hospitals, the vaccine or vaccine candidate was classified as “LE & U/P/H”. If developers included SMEs, LEs, and universities/public institutions/hospitals, the vaccine or vaccine candidate was classified as “SME & LE & U/P/H”.

## 3. Results

### 3.1. Development Status of Each Vaccine Platform

Approximately half of the COVID-19 vaccines (phase 4) and vaccine candidates (phases 1–3) used emerging vaccine platforms (Figure 1A). The most dominant vaccine platform used for COVID-19 vaccine development was the protein subunit, followed by RNA-based and viral vectors. Approximately 13% (24 of 182) of the COVID-19 vaccines and vaccine candidates used a conventional vaccine platform (inactivated/live-attenuated virus).

The clinical development stages of all the analyzed vaccines were classified into launch/late development (phases 2/3, 3, and 4) and early development (phases 1, 1/2, and 2) (Figure 1B). The older the platform used for vaccines and vaccine candidates, the closer the clinical development stage was to late development. Two-thirds of vaccines and vaccine candidates using conventional (inactivated/live-attenuated virus) vaccine platforms were approved or proceeded to late development, whereas 59–76% of vaccine candidates using emerging (viral vector, RNA-based, and DNA-based) vaccine platforms were still in early development.

### 3.2. Country Profile of COVID-19 Vaccine Developers

The country distribution of COVID-19 vaccine developers is shown in Figure 2. The country with the highest number of vaccine developers was China, followed by the U.S. Many COVID-19 vaccine developers were also from European countries. However, the number of developers in East Asia (Japan, South Korea, and Taiwan) was relatively low. In the regional classification of this study, South/Middle East/West Asia regions had a higher number of developers than East Asia. Vaccine developers were present in 32 countries. While traditional pharmaceutical development is predominantly led by the U.S., Europe, and Japan, COVID-19 vaccines have been developed in many other countries worldwide.

Next, the types of vaccine platforms developed in each country and region were examined (Table 1). In the U.S., China, Europe, and East Asia, conventional, established, and emerging vaccine platforms have been widely developed, while developers in South/Middle East/West Asia, Australia, and Latin America have been mainly developing conventional and established vaccine platforms. Among emerging vaccine platforms, RNA-based vaccines were most developed in the United States and China, viral vector vaccines in Europe, and DNA-based vaccines in East Asia.

### 3.3. Developer Profile of COVID-19 Vaccines by Vaccine Platform

The developers of COVID-19 vaccines were classified into four categories: SMEs founded in or after 2000 (representing startups); SMEs founded before 2000 (representing incumbent small pharma); LEs (representing incumbent large pharma); and universities/public institutions/hospitals (academia). The ratio of each category was then calculated in each vaccine platform (Figure 3). First, several universities/public institutions/hospitals were involved in developing COVID-19 vaccines on all vaccine platforms. This phenomenon was particularly prominent in inactivated/live-attenuated virus and viral vector vaccines, with universities/public institutions/hospitals representing 45% and 46% of the developers, respectively. Second, startups’ contributions as developers differed depending on the vaccine platform. Most companies that developed inactivated/live-attenuated virus vaccines were incumbent small pharma. In the established vaccine platforms (protein subunit, virus-like particle) and DNA-based vaccines, both startups and incumbent small pharma largely contributed to this development. In the two emerging vaccine platforms, viral vector and RNA-based, most companies that developed COVID-19 vaccines were startups, and more cases with the participation of incumbent large pharma in the development were observed than in other vaccine platforms.

**Figure 3 vaccines-12-00259-f003:**
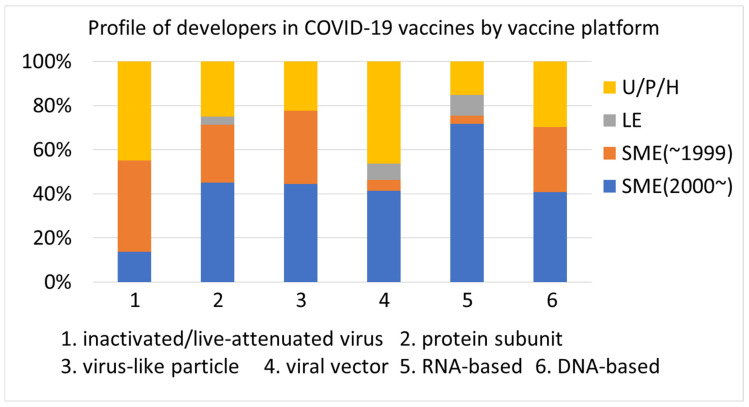
Developer profile of COVID-19 vaccines and vaccine candidates by vaccine platform. SME, small- and medium-sized enterprise; LE, large enterprise; U/P/H, university/public institution/hospital.

### 3.4. Country Distribution of Different Types of Developers

Figure 4 shows the country/region distribution of different types of developers (startups, incumbent small pharma, incumbent large pharma, and academia). Startups were primarily located in the U.S. and China. These two countries are leading in global unicorns [20] and a similar trend was observed in COVID-19 vaccine development. Most incumbent large pharma companies that engaged in COVID-19 vaccine development were located in Europe. In contrast, incumbent small pharma companies were more dominant in Asia and Latin America than in other countries and regions. Academia has contributed significantly to the development of COVID-19 vaccines worldwide.

**Figure 4 vaccines-12-00259-f004:**
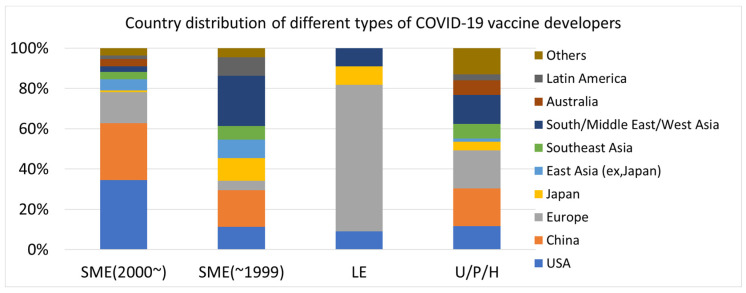
Country distribution of different types of COVID-19 vaccine developers. SME, small- and medium-sized enterprise; LE, large enterprise; U/P/H, university/public institution/hospital.

### 3.5. Alliance Situation in COVID-19 Vaccine Development

COVID-19 vaccines have been developed worldwide using various platforms, including emerging ones. Academia–industry and company-to-company alliances are important for expanding business overseas, leveraging new technologies, and acquiring new capabilities. Therefore, the alliance situation of COVID-19 vaccines and vaccine candidates was analyzed (Table 2). Among the COVID-19 vaccines and vaccine candidates in phases 2/3, 3, and 4, 35% of the development projects conducted alliances. Nearly half of the alliance projects were international (12 of 28). Regarding the patterns of alliance partners, SME and U/P/H were dominant (16 of 28), followed by SME and SME (5 of 28). Other patterns of alliance partners included only one or two cases. Among the COVID-19 vaccine candidates in phases 1, 1/2, and 2, 14% of the development projects conducted alliances. Half of the alliance projects were international (7 of 14). Regarding the patterns of alliance partners, SME and U/P/H were dominant (8 of 14), followed by SME and SME (4 of 14). The other alliance–partner patterns accounted for 0 or 1 case.

**Table 2 vaccines-12-00259-t002:** Alliance of developers in COVID-19 vaccines and vaccine candidates.

**Upper Panel: COVID-19 Vaccines and Vaccine Candidates in Phases 2/3, 3, 4**												
				**Alliance Type**		**Alliance Partner**						
**Vaccine Platform**	**No Alliance**	**Alliance**	**Alliance %**	**Domestic**	**International**	**SME & SME**	**LE & LE**	**U/P/H & U/P/H**	**SME & LE**	**SME & U/P/H**	**LE & U/P/H**	**SME & LE & U/P/H**
inactivated/live-attenuated virus	13	3	19%	0	3	1	0	1	0	1	0	0
protein subunit	18	10	36%	4	6	2	1	0	0	6	0	1
virus-like particle	3	0	0%	0	0	0	0	0	0	0	0	0
viral vector	6	7	54%	6	1	1	0	1	0	3	2	0
RNA-based	9	6	40%	5	1	1	0	0	1	4	0	0
DNA-based	2	2	50%	1	1	0	0	0	0	2	0	0
**Total**	**51**	**28**	**35%**	**16**	**12**	**5**	**1**	**2**	**1**	**16**	**2**	**1**
**Lower Panel: COVID-19 Vaccine Candidates in Phases 1, 1/2, 2**												
				**Alliance Type**		**Alliance Partner**						
**Vaccine Platform**	**No Alliance**	**Alliance**	**Alliance %**	**Domestic**	**International**	**SME & SME**	**LE & LE**	**U/P/H & U/P/H**	**SME & LE**	**SME & U/P/H**	**LE & U/P/H**	**SME & LE & U/P/H**
inactivated/live-attenuated virus	7	1	13%	0	1	0	0	0	0	1	0	0
protein subunit	28	3	10%	1	2	1	0	0	0	2	0	0
virus-like particle	3	1	25%	0	1	1	0	0	0	0	0	0
viral vector	17	2	11%	1	1	0	0	0	0	1	1	0
RNA-based	26	2	7%	1	1	1	0	0	1	0	0	0
DNA-based	8	5	38%	4	1	1	0	0	0	4	0	0
**Total**	**89**	**14**	**14%**	**7**	**7**	**4**	**0**	**0**	**1**	**8**	**1**	**0**

SME, small- and medium-sized enterprise; LE, large enterprise; U/P/H, university/public institution/hospital.

## 4. Discussion

Emerging vaccine platforms (viral vector, RNA-based, and DNA-based) had not been or were rarely applied to clinically used vaccines before the launched COVID-19 vaccines. Currently, these vaccine platforms are widely used for COVID-19 vaccines and vaccine candidates (Figure 1A), suggesting that these technologies have been rapidly immersed in the vaccine development space. Conventional (inactivated/live-attenuated virus) and established (mainly protein subunit) vaccine platforms are still widely used for COVID-19 vaccine development (Figure 1A), revealing that various vaccine platforms covering conventional, established, and emerging technologies are being utilized. Viral vector and RNA-based vaccine platforms have several advantages, such as a strong immune response without adjuvants [21,22], fast manufacturing, low cost [22,23], and rapid design [23,24]. However, these vaccine platforms are associated with concerns regarding side effects that are not observed in conventional and established vaccine platforms [24,25]. For future pandemics, it would be preferable to develop vaccines using various vaccine platforms while adopting a flexible approach, taking into consideration factors such as the speed of pathogen spread and the frequency of mutations.

Older vaccine platforms went into late development more often than newer vaccine platforms (Figure 1B). One may wonder if RNA-based and viral vector vaccines were the most advanced in development because the first three vaccines that were authorized and globally inoculated for COVID-19 used these two vaccine platforms. This analysis shows that, overall, vaccines and vaccine candidates using vaccine platforms that were more commonly used before the COVID-19 pandemic have advanced in clinical development. This is reasonable, assuming that older technologies are more sophisticated and require less optimization of development and manufacturing processes. Although emerging vaccine platforms are in clinical use for COVID-19 vaccines, developers require more time to use these technologies quickly and efficiently.

One noteworthy finding of this study was that China was the predominant country in which COVID-19 developers were present (Figure 2). According to a previous analysis, the number of drugs originating from China was quite small among the new drugs approved by the FDA over the past several years [10,13]. The presence of China has been rapidly increasing in COVID-19 vaccine development. China accounted for 27.2% of the most-cited papers published in 2018, 2019, and 2020 [26]. China’s world-class basic research capabilities are expected to directly contribute to its vaccine development. China has demonstrated robust research and development (R&D) capabilities in the field of COVID-19 vaccines and is anticipated to become one of the leading countries in global vaccine development in the near future. The U.S. has maintained its strength in vaccine development, considering the high number of developers of COVID-19 vaccines. In contrast, Japan has shown a relatively low presence of COVID-19 vaccine development. In this analysis, Japan accounted for only 10 COVID-19 vaccine developers, compared to 55 in China and 53 in the U.S. While Japan had the third largest pharmaceutical market in the world after the U.S. and China [27], it failed to rapidly develop COVID-19 vaccines. Japan’s pharmaceutical industry trade deficit further increased by approximately USD 7.5 billion in 2022 compared to 2021, which is largely attributed to the impact of vaccine imports [28]. The lack of a drug discovery ecosystem through venture entrepreneurship has weakened the international competitiveness of new drug development in the Japanese pharmaceutical industry [29]. In an era where the utilization of emerging vaccine platforms is expanding and diverse technological capabilities and speeds are required for vaccine development, it is unlikely that Japan will demonstrate a significant presence in global new vaccine development in the future. In contrast, these data demonstrate that COVID-19 vaccines have been developed in many pharmerging countries with emerging pharmaceutical markets, in addition to China. These countries did not have strong R&D capabilities for vaccines and drugs. Currently, the countries developing COVID-19 vaccines have expanded worldwide, including various Asian and Latin American countries, which means that vaccine platform technologies are available in several countries, and vaccine development is no longer exclusive to advanced pharmaceutical countries/regions such as the U.S., Europe, and Japan. Vaccine development has spread worldwide, and many countries are expected to contribute to the next pandemic vaccine development in the future.

Another interesting finding regarding country profiles was that the types of vaccine platforms varied by country or region. In the US, Europe, and East Asia, conventional, established, and emerging vaccine platforms were widely developed (Table 1), suggesting that various vaccine platforms have been available in the countries or regions with advanced pharmaceutical industries. Additionally, China has also developed various vaccine platforms, both old and new (Table 1). While China’s global presence in new drug development remains relatively low, it is considered that China already possesses the technological capability to accommodate diverse vaccine platforms. On the other hand, in other pharmerging countries (South/Middle East/West Asia, Australia, and Latin America), inactivated/live-attenuated virus and protein subunit vaccine platforms were mainly developed (Table 1). Most developers in these countries or regions are believed to have not yet established the emerging vaccine platform technologies. As mentioned above, in the future development of pandemic vaccines, utilizing both old and new vaccine platform technologies according to the situation is necessary. For the speedy utilization of vaccines worldwide, it might be effective for the US, Europe, China, and East Asia to develop vaccines using emerging vaccine platforms, while other pharmerging countries develop vaccines using conventional and established vaccine platforms. It was also an intriguing finding that differences in the utilization of emerging vaccine platforms were observed among countries or regions with advanced pharmaceutical industries. RNA-based vaccines have been predominantly developed in the U.S. and China, viral vector vaccines have been predominantly developed in Europe, and DNA-based vaccines have been predominantly developed in East Asia (Table 1). The reasons for this phenomenon are unclear. In future pandemic vaccine development, we may observe the utilization of vaccine platform technologies tailored to the characteristics of each country or region.

The contribution of academia (university/public institution/hospital) to COVID-19 vaccine development was observed on all vaccine platforms in varying degrees (Figure 3). The ratio of vaccines and vaccine candidates that academia was involved in developing was especially high for inactivated/live-attenuated virus (conventional vaccine platform) and viral vector (emerging vaccine platform), implying that academia participates in vaccine development regardless of whether the technology is old or new. The reason academia is prominent on these platforms is unclear. The importance of academia in vaccine development is well known [14,15] and academia will continue to play a major role in vaccine development using various vaccine platforms in future pandemic vaccine development. One viable option for vaccine development companies is to strengthen their collaboration with academia.

Conventional, established, and emerging vaccine platforms were developed mainly by incumbent pharma, both by incumbent pharma and startups, and mainly by startups, respectively (Figure 3). Startups play an especially important role in the development of vaccines using viral vector and RNA-based vaccine platforms, both of which are emerging (Figure 3). Startups, mainly university startups, play a crucial role in translating cutting-edge medical technologies into clinical applications [9]. The government should implement startup promotion measures to increase the capability of novel vaccine development. Large pharma companies are important for shepherding new pharmaceutics discovered by startups through clinical development and approval [30]. The participation of large pharma companies was observed in some cases of viral vector and RNA-based vaccines in this analysis, demonstrating that both startups and large pharma have actively engaged in the development of vaccines using these emerging vaccine platforms. In COVID-19 vaccines, clinical development of an adenoviral vector vaccine discovered by the University of Oxford was supported by AstraZeneca and Moderna received large collaboration fees from some large pharma companies, which supported Moderna’s long-term R&D efforts [31]. Strengthening the alliance between startups and large pharma can be an option to increase the efficiency of clinical applications of new vaccine technologies in the future.

Startups involved in COVID-19 vaccine development are dominant in the U.S. and China, and many other countries rely on incumbent pharma (mostly small pharma in pharmerging countries) for vaccine development (Figure 4). Each country has its own national innovation system, which determines the innovation characteristics and performance of the nation [17]. For instance, in the U.S., the government’s industrial policy, such as Small Business Innovation Research, has enhanced the establishment of university startups, and this has led to the development of new products, including pharmaceuticals [32]. Investment in startups has been active, and many “unicorn” startups have been generated in the U.S. [20]. In contrast, the R&D activities of large incumbent companies have led to innovation in Japan. Therefore, science-based industries that utilize basic academic research outcomes have shown weak international competitiveness [28]. The Japanese pharmaceutical industry has been increasing its trade deficit for at least the past thirty years [28]. Startup ecosystems in Japan have not been nurtured, and the amount of investment in startups in Japan is only one-hundredth of that in the U.S. [33]. Various political measures have been implemented to generate startups and improve industrial competitiveness in Japan [34]. Thus, by analyzing a country’s national innovation system and its differences, it is possible to predict its innovation performance. This, in turn, is expected to contribute to the country’s science and technology policies and corporate technological strategies. Startups have greatly contributed to new drug discovery in the U.S., hybrid contributions by startups and incumbent pharma have been observed in Europe, and incumbent pharma is dominant in Japan [29]. A similar trend was observed in COVID-19 vaccine development. In addition, Chinese startups have been highly engaged in the development of COVID-19 vaccines. Startups have made significant contributions to the development of vaccines using emerging vaccine platforms. Considering this, it is likely that the U.S. and China will lead the development of novel vaccine platforms. COVID-19 vaccine development has also been conducted in many pharmerging countries, and the main developers in these countries are incumbent small pharma and academia (Figure 4). They should continue developing vaccines using conventional and established vaccine platforms, and establish robust startup ecosystems to develop vaccines that incorporate more advanced technologies in the future.

Among the COVID-19 vaccines and vaccine candidates in the phase of launch/late development (phases 2/3, 3, and 4), 35% of the development projects involved alliances (Table 2, upper panel). This ratio was only 14% for projects under early development (phases 1, 1/2, and 2) (Table 2, lower panel). This study did not analyze the contents of alliances; therefore, concrete conclusions cannot be drawn. However, it is implied that developers mainly utilize external capabilities and funds that are required for large clinical trials and manufacturing rather than technologies to design and create vaccine candidates; otherwise, they would collaborate in earlier phases. Nearly half of the alliances were international, in both the early and late development phases, implying that international players were actively involved in development alliances. Considering that developers from 32 countries participated in the development of COVID-19 vaccines, it can be inferred that several countries are now acquiring vaccine development capabilities and cooperating in clinical development. International collaboration in vaccine development is anticipated to continue advancing. Regarding alliance partners, alliances between SMEs and academia were the most apparent (Table 2), implying that the cutting-edge technologies of startups and the clinical capability of academia are likely to be matched, although the contents of alliances were not analyzed in this study. Interestingly, the second most dominant pattern of alliances in COVID-19 vaccine development was SME-to-SME collaboration (Table 2). Recently, SMEs have expanded their role in new drug discovery from the discovery of highly novel drugs to late-entry-improved drugs [13]. They have also extended their participation in late clinical development by boosting startup funding [35]. Considering the increasing presence of startups in drug discovery and development, it is anticipated that by boosting collaboration, startups will further increase their contributions to new vaccine development.

## 5. Conclusions

Vaccine platforms and developers in COVID-19 vaccine development as of March 2023 were comprehensively analyzed. Vaccines and vaccine candidates have been developed worldwide using various vaccine platforms, including conventional, established, and emerging platforms. The country distribution of developers reveals that China and the U.S. dominate in COVID-19 vaccine development worldwide, followed by Europe. In contrast, Japan, one of the countries with advanced pharmaceutical industries, has reduced its presence in vaccine development. Instead, many pharmerging countries possess the capability to develop vaccines. The important role of academia in vaccine development, which has long been viewed, still remains in COVID-19 vaccines. It will be continuously important to seek academia-industry collaboration in future pandemic vaccine development. Emerging vaccine platforms have been developed mainly by startups in COVID-19 vaccine development. It is important to enhance the establishment of a startup ecosystem to accelerate novel vaccine platform technologies. The startups are dominated in the U.S. and China, anticipating their leading role in the development of novel vaccine platforms in the future. In the development of COVID-19 vaccines, alliances have progressed to later clinical stages, with numerous international collaborations. Pandemic vaccine development using various platforms is expected to be actively pursued worldwide by leveraging international alliances.

One limitation of this study was that it focused only on COVID-19 vaccine development. In the case of COVID-19, national support for vaccine development and global efforts have been promoted in response to this unprecedented pandemic. The development of pandemic vaccines can be influenced by various factors such as the origin of the virus, affected regions, rate of spread, scale of infection, mortality rate, propensity for mutation, and so on. Additionally, different viruses may exhibit variations in the ease of vaccine production across various vaccine platform technologies. Therefore, it is not certain whether observations regarding COVID-19 vaccines can be applied to other pandemic vaccine developments in the future. In this study, spatial correlation analysis was not conducted. The topic of how the technological and developmental trends in one region influence the surrounding areas is intriguing and could become a future research topic. Even considering these limitations, it is believed that a comprehensive clarification of the current global status of COVID-19 vaccine development will not only provide important information for predicting trends in vaccine development in future pandemics but will also provide significant suggestions for national policy development and company strategies.

## Figures and Tables

**Figure 1 vaccines-12-00259-f001:**
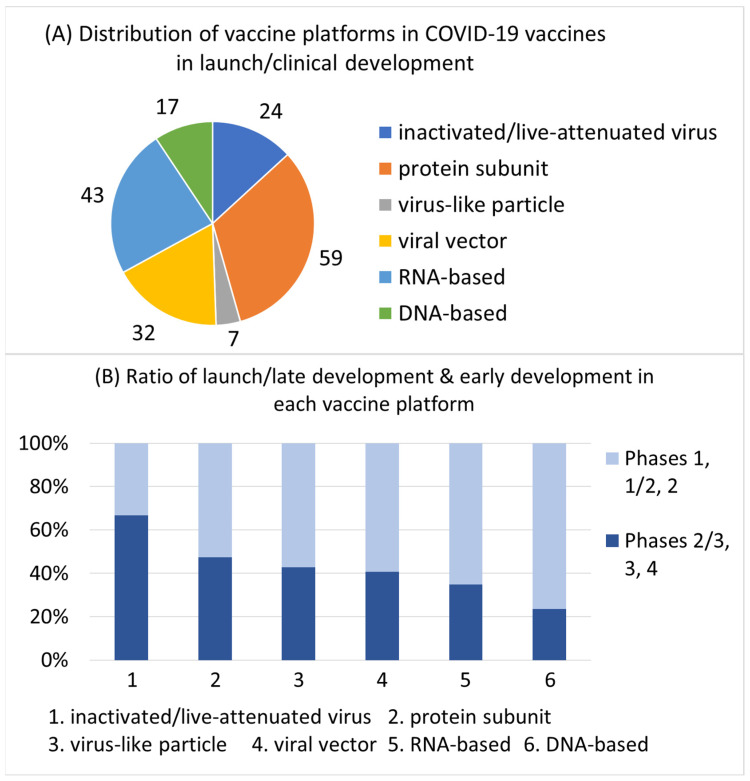
Analysis of vaccine platforms used for COVID-19 vaccines and vaccine candidates. (**A**) Distribution of vaccine platforms in COVID-19 vaccines and vaccine candidates. (**B**) Ratio of launch/late clinical development and early clinical development by vaccine platform.

**Figure 2 vaccines-12-00259-f002:**
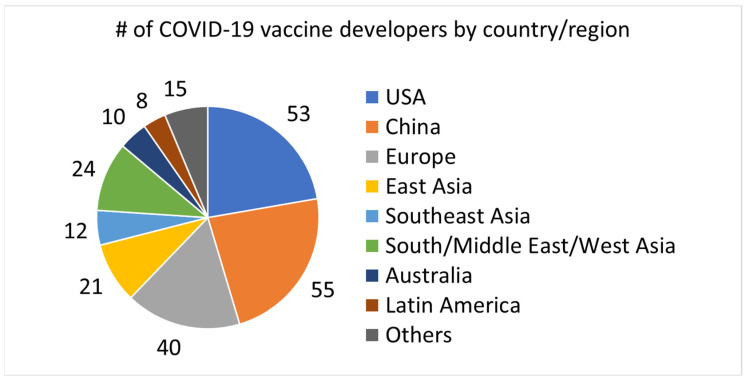
Country distribution of COVID-19 vaccine developers.

**Table 1 vaccines-12-00259-t001:** Number of COVID-19 vaccine developers using each vaccine platform in each country/region.

	Vaccine Platform					
Country/Region	Inactivated/Live-Attenuated Virus	Protein Subunit	Virus-like Particle	Viral Vector	RNA-Based	DNA-Based
USA	4	16	2	8	20	3
China	6	19	1	11	15	3
Europe	2	11	2	11	7	7
East Asia	2	7	0	1	3	8
Southeast Asia	2	5	0	2	2	1
South/Middle East/West Asia	9	6	2	3	2	2
Australia	0	6	1	0	1	2
Latin America	1	6	0	0	1	0
Others	3	4	1	4	2	1

## Data Availability

The data that support the findings of this study are available from the corresponding author upon reasonable request.

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
