# Peer review of "Trends in COVID-19 Vaccine Development: Vaccine Platform, Developer, and Nationality"

_vaccines, 2024, doi:10.3390/vaccines12030259_

Round 1
Reviewer 1 Report
Comments and Suggestions for Authors
The manuscript titled "Trends in COVID-19 vaccine development: vaccine platform, developer, and nationality" by Okuyama et al.analyzed vaccine platforms and developers involved in COVID-19 vaccine development, elucidating the trends of vaccine platforms used, the country distribution of the developers, and differences in the profiles of developers by vaccine platform technologies and country. The article is well-structured, to improve the current version of the manuscript, my comments are described as below.
1. Enhancing the clarity of these figures. An orderly and aesthetically pleasing arrangement of images will help readers better understand the content of the article.
2. The content of the conclusion is too redundant, please refine the conclusion to succinctly summarize the key findings, their significance, and potential areas for future research.
3. While the article provides a comprehensive overview of vaccine development trends, a more detailed comparative analysis between different countries and vaccine platforms could strengthen the paper. This could include a discussion on the implications of these trends for future pandemic preparedness.
4. Expand the discussion on the limitations of the study. While the article mentions the focus on COVID-19 vaccine development, further elaboration on the implications of these limitations for the study's findings would be beneficial.
5. Please ensure that all references are up-to-date and accurately cited.
Comments on the Quality of English LanguageMinor editing of English language required
Reviewer 2 Report
Comments and Suggestions for Authors
The manuscript under review is an interesting paper that analyses and describes the recent trends in COVID-19 vaccines development. The paper is well-written and clear. The methods are adeguate to the type of study. I congrant with the author and only have a few minor comments :
- I would avoid the use of the first person "I" in a scientific paper, expecially when describing the methods or exposeing the results. So, where it is possibile, I suggest to modify these type of sentences.
- if the author agrees, I suggest to divide the methods part in subparagraph, to make it clearer.
- the conclusions are very too long, I suggest to move some part of the conclusions into the discussion or remove some sentences. The conclusions should be the very summary of the conclusions of the discussion.
Reviewer 3 Report
Comments and Suggestions for Authors
This is a well written paper
There needs to be a better presentation of the research methodology
Did the author check for spatial correlation? This needs to be done.
It appears that this is just a presentation of facts of the SMEs. That is fine, but that is not sufficient enough. A research methodology and presentation of the results is necessary. How does location impact the vaccines, what about public investment, how does this impact vaccine development?
Comments on the Quality of English Languageminor editing
